# Isolation and Characterization of Galloylglucoses Effective against Multidrug-Resistant Strains of *Escherichia coli* and *Klebsiella pneumoniae*

**DOI:** 10.3390/molecules27155045

**Published:** 2022-08-08

**Authors:** Nelson E. Masota, Knut Ohlsen, Curd Schollmayer, Lorenz Meinel, Ulrike Holzgrabe

**Affiliations:** 1Institute of Pharmacy and Food Chemistry, University of Wuerzburg, Sandering 2, 97074 Wuerzburg, Germany; 2School of Pharmacy, Muhimbili University of Health and Allied Sciences, Upanga West, Dar es Salaam P.O. Box 65013, Tanzania; 3Institute for Molecular Infection Biology, University of Wuerzburg, Josef-Schneider-Strasse 2, 97080 Wuerzburg, Germany

**Keywords:** antimicrobial resistance, *Escherichia coli*, *Klebsiella pneumoniae*, Enterobacteriaceae, Paeonia, gallotannins, isolation, structural elucidation

## Abstract

The search for new antibiotics against multidrug-resistant (MDR), Gram-negative bacteria is crucial with respect to filling the antibiotics development pipeline, which is subject to a critical shortage of novel molecules. Screening of natural products is a promising approach for identifying antimicrobial compounds hosting a higher degree of novelty. Here, we report the isolation and characterization of four galloylglucoses active against different MDR strains of *Escherichia coli* and *Klebsiella pneumoniae*. A crude acetone extract was prepared from *Paeonia officinalis* Linnaeus leaves, and bioautography-guided isolation of active compounds from the extract was performed by liquid–liquid extraction, as well as open column, flash, and preparative chromatographic methods. Isolated active compounds were characterized and elucidated by a combination of spectroscopic and spectrometric techniques. In vitro antimicrobial susceptibility testing was carried out on *E. coli* and *K. pneumoniae* using 2 reference strains and 13 strains hosting a wide range of MDR phenotypes. Furthermore, in vivo antibacterial activities were assessed using *Galleria mellonella* larvae, and compounds 1,2,3,4,6-penta-O-galloyl-β-d-glucose, 3-O-digalloyl-1,2,4,6-tetra-O-galloyl-β-d-glucose, 6-O-digalloyl-1,2,3,4-tetra-O-galloyl-β-d-glucose, and 3,6-bis-O-digalloyl-1,2,4-tri-O-galloyl-β-d-glucose were isolated and characterized. They showed minimum inhibitory concentration (MIC) values in the range of 2–256 µg/mL across tested bacterial strains. These findings have added to the number of known galloylglucoses from *P. officinalis* and highlight their potential against MDR Gram-negative bacteria.

## 1. Introduction

Due to the ongoing global threat of antimicrobial resistance, the search for novel antimicrobial agents is crucial. Approaches aimed at the discovery and development of novel antibiotics are key with respect to continuously filling the antibiotics development pipeline. Many approaches have been applied in the search for new antibiotics, including modification of existing antibiotics, combination therapies, the use of resistance modifiers, as well as searching for novel antibiotics from scratch or from natural sources [1,2].

Antimicrobial resistance among Gram-negative bacteria poses a serious public health threat, as such infections are generally difficult to treat. *Escherichia coli* and *Klebsiella pneumoniae*, among other Gram-negative bacteria, are categorized by the World Health Organization (WHO) as high-priority pathogens against which the development of new therapies is vital. This is mainly due to their increasing resistance to carbapenems, which has narrowed the available treatment options. The use of the drug of last resort (colistin) is also challenged by the rise of plasmid-mediated transfer of the mcr-1 colistin resistance gene [3,4].

To significantly tackle the antimicrobial resistance (AMR) crisis, antibiotics with novel chemical structures and novel targets that act by new modes of action and lack cross resistance to existing antibiotics are urgently needed. However, most candidates currently in the pipeline fall short of these noble traits, which necessitates continual efforts in the search for ideal candidates [2].

Natural sources, especially from the Fungi and Monera kingdoms, have long been a reliable source of a number of available antibiotics. Due to their structural novelty and complexity, nature-derived compounds have formed successful classes of antibiotics, enabling the realization of new targets and modes of action [5]. Nevertheless, the role played by the driving factors for the development and spread of AMR has rendered most natural compounds less useful. Plants remain an attractive source for the discovery of new antibiotics due to their potential to host compounds with a higher degree of structural novelty, among other features of ideal new antibiotics [6]. 

*Paeonia officinalis* Linnaeus. (Paeoniaceae) is a small, non-bushy flowering plant naturally distributed in North America, Asia, and southern Europe [7]. Preparations from different parts of the plant are traditionally used to treat a broad range of diseases, including liver diseases, epilepsy, infections, pain, gastritis, amenorrhea, dysmenorrhea, as well as treatment of wounds, among other applications [7]. Flavonoids, tannins, glucosides, triterpenoids, phenols, and stilbenes are among the compounds previously isolated from this genus. Numerous galloylglucoses, among other tannins, were previously isolated from the leaves, fruits, petals, and roots of several peonies [7]. The available toxicological data favor plants from the genus Paeonia as largely safe. However, compared to other species from the genus, the availability of data from *P. officinalis* on isolated compounds and their biological activities is scarce [8,9].

A number of tannins were previously isolated from different parts of *P. officinalis* and are typically classified into hydrolysable, condensed, and complex tannins. Tannins play important roles in plant growth, reproduction, and natural defense systems. In addition to providing a chemical barrier against the penetration and colonization of plant tissues by microorganisms, they help to deter predation by herbivores and insects [10,11,12]. Galloylglucoses, as hydrolysable tannins, are biosynthesized in plants following a strictly position-specific series of galloylation of glucose [10,13]. In addition to antibacterial activities, galloylglucoses are reported to exhibit nematicidal, insecticidal, antioxidant, anti-inflammatory, antidiabetic, cardioprotective, neuroprotective, anticancer, antiplatelet, anticoagulant, and immune-modulatory effects, among other biological activities [12,13,14,15,16].

Our previous efforts in the search for antibacterial compounds from numerous plants revealed antibacterial activity of crude acetone leaf extract of *P. officinalis* against *E. coli* and *K. pneumoniae* [17]. Here, we report the isolation, purification, and characterization of four galloylglucoses from the leaves of *P. officinalis* and their antibacterial activities against 2 reference and 13 multidrug-resistant (MDR) strains of *E. coli* and *K. pneumoniae*.

## 2. Materials and Methods

### 2.1. Chemical Reagents and Antibiotics

Acetone, n-hexane, toluene, ethyl acetate, dichloromethane, methanol, acetonitrile, silica gel 60 GF254 plates, and silica gel 60 powder were purchased from Sigma-Aldrich Chemie (Schnelldorf, Germany). Mueller–Hinton broth (MHB), Lysogeny broth (LB), and agar were purchased from Carl Roth (Karlsruhe, Germany). Gentamicin sulfate was purchased from AppliChem (Darmstadt, Germany). Phosphate-buffered saline (PBS) pH 7.4 was purchased from Fischer Scientific (Schwerte, Germany) and Millipore water was prepared by the Milli-Q^®^ direct laboratory water purification system (Merk, Darmstadt, Germany).

### 2.2. Study Organisms

Bacterial reference strains of *Escherichia coli* (ATCC 25922) and *Klebsiella pneumoniae* (ATCC 10031) were purchased from the American Type Culture Collection (ATCC-LGC, Wesel, Germany). MDR Strains with the following identification numbers and resistance phenotypes in the parentheses were obtained from the Institute of Hygiene and Microbiology of the University of Wuerzburg (Wuerzburg, Germany): *E. coli*: NRZG 176 (OXA-48), Stich E 866 (VIM-1), UR481/1/2/13 (ESBL), NRZG 222 (KPC-3), RV3/A2/12 (VIM-1/4, TEM-1), and NRZG 14,408 (KPC-2, mcr-1); and *K. pneumoniae*: NRZG 246 (OXA 48), Stich E 895 (TEM/SHV/CTX-M), UR 3397/1/15 (NDM-1), Stich 787 (DHA-1 (AmpC)), NRZG 103 (KPC-2), NRZG 002 (OXA-48), and Stich E 785 (SHV-4). Moreover, *Galleria mellonella* larvae were purchased from Feeders & More GmbH (Au in der Hallertau, Germany).

### 2.3. Plant Material Collection, Preparation, and Extraction

Leaves of flowering *Paeonia officinalis* L. plants were collected from the Botanical Garden of the University of Wuerzburg, Germany in August 2019. Plant identification was carried out by a botanist (Dr. Gerd Vogg) and was assigned internal ascension number 2013-11-S-10. The collected leaves were then transported to the laboratory in aerated paper bags, where they were chopped into small pieces and dried under shade and open air for two weeks. Dried leaves were thereafter reduced into a coarse powder using an electric blender (Braun, Kronberg in Taunus, Germany).

Extraction was performed under cold maceration using 44 g of powdered dry leaves and acetone at a solvent with a feed ratio of 10 mL/g for 72 h under constant magnetic stirring. The contents were then filtered, and the solvent was evaporated in vacuo at 40 °C. The recovered crude extract (4.58 g, 10.4% *w*/*w*) was stored at −15 °C.

### 2.4. Isolation and Purification

Bioassay-guided isolation was performed after the identification of the antibacterial activities of the crude extract on the reference bacterial strains. A contact bioautography technique was used to guide the identification of spots with antibacterial activity on developed thin-layer chromatography (TLC) plates.

#### 2.4.1. Contact Bioautography

A mobile phase containing n-hexane: ethyl acetate: methanol (7.5:2:0.5 *v*/*v*/*v*) resulted in the largest number of distinct spots on a precoated silica gel 60 GF254 plate. To develop a TLC profile, 20 µL of a 10 mg/mL crude extract solution in acetone was spotted at two separate positions 1 cm from the bottom of an 8 cm × 2.5 cm TLC plate. The plates were developed using the above mobile phase and allowed to dry for 6 h in a sterile hood. Another control plate was developed in a similar way, and the position of each spot was marked under ultraviolet (UV) light at 254 and 366 nm. The control plate served as a template for the identification of the position, size, and intensity of the bioactive spots. A plate spotted with 20 µL of acetone with similar subsequent treatments served as a negative control [18].

A volume of 100 µL of bacterial suspensions with 10^8^ colony-forming units (CFU)/mL prepared from fresh cultures was inoculated and uniformly spread on Mueller–Hinton agar plates. The inoculated plates (one for each bacterium and one for control) were lid-covered and left for 30 min at room temperature. The developed TLC described above was thereafter placed on the surface of the inoculated agar plate (the silica-coated surface facing the agar) and gently pressed to ensure uniform contact. The position of the TLC plate on the agar plate was traced/marked from the agar side of the agar plate’s exterior. The entire set was kept at 4 °C for 30 min to facilitate the diffusion of the compounds into the agar. The TLC plate was thereafter carefully removed under sterile conditions, and the agar plates were incubated at 37 °C for 24 h. The spots containing compounds with antibacterial activity were identified by matching the zones of inhibition to the corresponding position on the control TLC plate [18].

#### 2.4.2. Liquid–Liquid Extraction

Guided by the results of contact bioautography, liquid–liquid extraction was carried out to simplify the crude extract’s composition. The extraction was performed as per a previously described procedure, with slight modifications [19]. A total of 4 g of the crude extract was dissolved into 200 mL of 90% methanol and extracted with three 100 mL portions of petroleum ether using a separating funnel. Both layers were dried in vacuo at 40 °C, and the methanolic extract was resuspended in 200 mL of water and extracted with three 100 mL portions of chloroform. Combined chloroform layers were dried in vacuo at 40 °C, whereas the water layer was freeze-dried (Martin Christ Gefriertrocknungsanlagen, Osterode am Harz, Germany) for 24 h to recover 1.92 g (48%) of a dark-brownish powder. Dried subfractions from petroleum ether, chloroform, and water, were tested for antibacterial activity using disc diffusion assays on Mueller–Hinton agar. The optimum mobile phase composition of dichloromethane: toluene: methanol (20:35:45% *v*/*v*/v) was thereafter used to develop the TLC plates for bioautography assay of the recovered dried water fraction. The results from bioautography and the developed TLC profile guided the subsequent isolation step by open-column chromatography.

#### 2.4.3. Open-Column and Flash Chromatography

The dried water subfraction (1.92 g) from liquid–liquid extraction was subjected to open-column subfractionation using silica gel 60 and dichloromethane: toluene: methanol (20:35:45% *v*/*v*/v) as a starting mobile phase system, followed by methanol: water (95:5% *v*/*v*). The fractions containing the spot with the active compound(s) (corresponding to a spot with an Rf value of 0.54) were pooled and dried in vacuo at 40 °C.

Further refining of the resulting subfraction was performed by flash chromatography (Interchim PuriFlash 430 Flash Chromatography System, France) using a reversed-phase column (Chromabond^®^ Flash RS 40 C_18_ ec, Macherey-Nagel, Dueren, Germany) eluted with n-hexane: ethyl acetate: water (25: 65:10% *v*/*v*/v) under isocratic conditions. The classic reversed-phase chromatography (methanol or acetonitrile-water) systems could not yield optimal results. Fractions containing a spot corresponding to the active compounds at Rf = 0.72 on reversed-phase, pre-coated silica gel C_18_ TLC plates (ALUGRAM^®^ RP-18W/UV_254_, Macherey-Nagel, Dueren, Germany) developed with the above mobile phase were pooled and dried in vacuo at 40 °C followed by freeze drying for 24 h to recover 656.6 mg (32.4%) of light-brownish amorphous powder.

#### 2.4.4. Preparative Reversed-Phase High-Performance Liquid Chromatography (RP-HPLC)

Reversed-phase HPLC analysis of the refined water subfraction revealed the presence of multiple components, which necessitated further separation under preparative HPLC (Agilent 1100 Series Preparative HPLC, Santa clara, CA, US). Isolation of subfractions from the above fraction was carried out under conditions indicated in Appendix A.

Purified subfractions/compounds were dried in vacuo at 40 °C, followed by freeze-drying for 24 h. The recovered compounds were stored at −15 °C until they were needed for further experiments. Bioactivities of each recovered subfraction (compound) were tested using either broth microdilution assay or disc diffusion assay in cases where only small amounts were recovered.

### 2.5. Characterization and Structural Elucidation

#### 2.5.1. Infrared Spectral Measurements

Infrared spectra of the active purified compounds were acquired using a JASCO FT/IR-4700 Fourier transform infrared (FT-IR) spectrometer equipped with an attenuated total reflectance (ATR) accessory (JASCO Labor und Datentechnik GmbH, Pfungstadt, Germany).

#### 2.5.2. High-Resolution Mass Spectrometry

Exact masses of the active purified compounds were obtained with an Exactive™ Plus Orbitrap high-resolution mass spectrometer (HR-ESI-MS) (ThermoFischer Scientific, Waltham, MA, US) under electrospray ionization in both positive and negative modes. Moreover, full mass spectra, simulated spectra, and calculated masses for the proposed molecular formulae were also acquired/processed.

The identity of the measured compounds was partly confirmed by the screening of relevant databases (MassBank Europe, PubChem, ChemSpider, and SciFinder) and available literature for compounds with similar exact masses and proposed molecular formulae.

#### 2.5.3. Nuclear Magnetic Resonance

One-dimensional (^1^H, ^13^C, and distortionless enhancement by polarization transfer-135 (DEPT-135)) and two-dimensional (COrrelation SpectroscopY (COSY), heteronuclear single-quantum correlation (HSQC), and heteronuclear multiple-bond correlation (HMBC)) 600 MHz NMR measurements of the active purified compounds were carried out on a Bruker Avance III HD 600 MHz NMR spectrometer (Bruker, Billerica, MA, US). All HMBC measurements were done at the long-range (2–3 bonds) J(CH) coupling constant of 8.3 Hz. Based on differences in their solubilities, the compounds were dissolved in different proportions of acetone-d6 in D_2_O. Compounds NMA2 (21 mg/mL), NMB4 (8.3 mg/mL), NMB6 (25 mg/mL), and NMC3 (15.4 mg/mL) were dissolved in 91, 75, 100 and 96.2% of acetone-d6 in D_2_O (*v*/*v*), respectively [20,21,22,23,24,25].

### 2.6. Antibacterial Activity Testing

#### 2.6.1. Disc Diffusion Assays

A disc diffusion assay was used to evaluate antibacterial activities of fractions isolated in small quantities before deciding on their further isolation or structural elucidation. All assays were conducted as per the guidelines of the European Committee for Antimicrobial Susceptibility Testing (EUCAST), with slight modifications. Briefly, Mueller–Hinton agar plates were inoculated with 100 µL of 10^8^ CFU/mL bacterial suspensions and left under sterile conditions for 30 min at room temperature. Then, 6 mm diameter test discs were loaded with an amount equivalent to 256 µg of the test substance solution dissolved in acetone. The discs were left to dry for 1h in a biosafety hood before being transferred to the inoculated agar plates mentioned above. Discs loaded with acetone alone were used as negative controls. Zones of inhibition were read after 24 h of incubation at 37 °C [26].

#### 2.6.2. Broth Microdilution Assays for Determination of Minimum Inhibitory Concentration (MIC) 

##### Preparation of Bacterial Suspensions

To prepare overnight cultures, one isolated colony of each bacterium was picked up using a sterile tip and used to inoculate 2 mL of lysogeny broth (LB) medium in sterile test tubes. The tubes were then incubated overnight (12–14 h) at 37 °C under constant shaking (200 rpm). Fresh cultures were thereafter prepared by transferring 200 µL of the overnight cultures into 20 mL of LB medium in sterile flasks and further incubated for 5–6 h under the same conditions as above. Volumes of fresh cultures needed to prepare final test bacterial suspensions were determined using the optical density (OD_600_) (Eppendorf BioPhotometer Plus, Eppendorf, Hamburg, Germany) of the respective fresh culture and the Newman’s correlation curve. All test bacterial suspensions were prepared at concentrations of 10^6^ CFU/mL [27].

##### Loading of 96-Well Plates

MICs of the crude extract, fractions, and purified compounds were determined by the broth microdilution method as per EUCAST guidelines. Stock solutions of the crude extract and petroleum ether, chloroform and water subfractions were prepared at 10 mg/mL, whereas those of purified compounds were prepared at 2 mg/mL in acetone. Through dilution with MHB media, working solutions at concentrations of 2048 µg/mL and 512 µg/mL were made from the above stock solutions. The concentration of acetone in the working solution was 25.6% *v*/*v*.

To ensure a uniform concentration of acetone across all wells, 100 µL of 25.6% *v*/*v* acetone in MHB medium was prefilled in wells on columns 3 to 11 of the 96-well plate in triplicate. This was followed by loading 200 µL of the working solutions into the wells on column 2 of the corresponding prefilled triplicate rows. The working solution was serially diluted using a multichannel pipette by drawing 100 µL of the working solution and mixing it well with the above prefilled 100 µL acetone-MHB mixture in the successive columns. The procedure was repeated until the 11th column was reached, at which point the final 100 µL was discarded.

To the above wells, 100 µL of bacterial suspension in MHB medium at 10^6^ CFU/mL were added, attaining test concentration ranges of 2–1024 μg/mL for the crude extract and 0.5–256 μg/mL for subfractions/compounds. Furthermore, a final concentration of 12.8% *v*/*v* acetone in MHB medium was achieved across all test wells. The wells on the outer ring of the plate were filled with 200 µL of MHB medium, which reduced the evaporation of acetone from the treatment and control wells within. Gentamicin sulfate was used as a positive control in the range of 0.25–128 μg/mL µg/mL, whereas a 12.8% *v*/*v* of acetone in MHB medium mixture was used as a negative control. The loaded and inoculated plates were thereafter incubated at 37 °C for 18 h. All experiments were performed in triplicate in one assay and repeated three times on separate days. MIC values were determined by visual observation for the absence of pinpointed bacterial growths at the bottom of treatment wells. MIC values were recorded as the highest of the values obtained from all individual replicas [27,28].

#### 2.6.3. Minimum Bactericidal Concentration (MBC) Testing

Minimum bactericidal concentration (MBC) values were determined as per the EUCAST guidelines. Briefly, after 18 h of incubation for determination of MIC values as described above, 20 µL was drawn from the test wells without bacterial growth (MIC and above) and applied to the MHB agar plates in triplicate. The plates were then incubated for a further 24 h at 37 °C. The MBC was determined, as the wells with the lowest concentration with no bacterial growth (colonies) were observed at the end of the incubation time.

### 2.7. In Vivo Antibacterial Assay Using Galleria mellonella Larvae

In vivo antibacterial assays were performed using *Galleria mellonella* larvae. To avoid maturation and pupation, all larvae were used within 4 days after their delivery. Bacterial suspensions of the reference strains (*E. coli* (ATCC 25922) and *K. pneumoniae* (ATCC 10031)) to infect the larvae were prepared from overnight and fresh cultures using similar procedures as those described above. Moreover, to remove residual bacterial toxins, the fresh cultures were centrifuged under mild conditions (3500 rcf for 5 min at 4 °C), and the supernatant was discarded, followed by resuspension of the bacteria in PBS. The resulting suspension was recentrifuged under the same conditions, and the bacteria were resuspended in PBS after careful removal of the supernatant. The optical density (OD_600_) of the resulting bacterial suspension in PBS was measured, and using Newman’s correlation curves, the desired concentrations of bacterial suspensions in PBS were prepared for inoculation of the *G. mellonella* larvae.

The optimal bacterial concentrations to be used for infection of the larvae before treatments were identified after testing the doses between 0.05 × 10^8^–3.0 × 10^8^ CFU/mL for *E. coli* (ATCC 25922) and *K. pneumoniae* (ATCC 10031). The selected optimal concentration was that which did not kill more than 50% of the untreated infected larvae within 12 h of incubation while ensuring that 80% or more of the untreated larvae were killed within 24 h of incubation.

Larvae were infected by injecting 20 µL of the bacterial suspension on the second last appendage on the left-hand side of each larva using a BD MicroFine + Demi 0.3 mL, 0.30 mm (30 G) × 8 mm insulin syringe (BD, Wokingham, UK). A total of 20 larvae were used per test group. Each group was placed on plastic Petri dish plates with a small amount of wooden chips. The infected larvae were thereafter incubated at 37 °C for 30 min before they were treated with the test compounds and controls [28].

Larvae were treated with the test compounds and controls by injecting 20 µL of each test compound in 12.8% acetone in PBS on the same appendage as described above. Compounds were tested at fixed concentrations of 64 or 128 μg/mL, which were generally two times their in vitro MIC values. The larvae were then incubated at 37 °C, and dead larvae were counted every 6 h for the first 48 h and then every 12 h until 96 h after treatment. Dead larvae were identified by the virtue of a complete lack of response to touch stimuli and were removed from the plates [28]. The negative control group was treated with 20 µL of 12.8% acetone in PBS, whereas the positive control was given the same volume of gentamicin 64 µL/mL solution in PBS. Furthermore, one group without any treatment was included.

### 2.8. Evaluation of the Amino Acid Composition of the Expressed Resistance Enzymes

The literature broadly indicates stronger interactions of galloylglucoses with proteins with higher contents of aromatic amino acids, as well as their electrostatic adsorption onto the surfaces of positively charged moieties [29,30,31,32,33,34]. On these grounds, we evaluated the relationships between the observed MIC values and the contents of aromatic amino acid, as well as net charges in the resistance enzymes in the studied MDR bacteria. Sequences of the amino acids in each resistance enzyme were obtained from the Protein Data Bank (PDB) and UniProt databases [35,36], and their relative contents of aromatic amino acids and net charges were determined using an online protein analysis tool (ExPASy) and Origin^®^ software [37]. To determine possible relationship patterns, the above parameters were scatter-plotted against the MIC values observed among studied MDR strains hosting the indicated resistance enzymes. The analysis considered only cases in which the MIC values within our highest tested concentration of 256 μg/mL were obtained.

## 3. Results

### 3.1. Extraction, Isolation, and Purification of Antibacterial Compounds

Crude acetone leaf extract from *P. officinalis* dried leaves was obtained with a 10.4% *w*/*w* yield after 72 h of cold maceration. The extract showed moderate antibacterial activities (MIC = 128–258 µg/mL) against *E. coli* (ATCC 25922) and *K. pneumoniae* (ATCC 130031) (Figure 1). Upon TLC profiling of the extract (n-hexane: ethyl acetate: methanol (7.5:2:0.5 *v*/*v*/v)), multiple spots (compounds) were visible across the entire run distance. Contact bioautography on the TLC plate developed under the above conditions exhibited a zone of inhibition at the position of sample application (Rf = 0). These findings suggested a high polarity of the antibacterial compound(s) present in the crude extract, which informed the decision to conduct liquid–liquid extraction to simplify the extract. As shown in Figure 1, the aqueous fraction from liquid–liquid extraction was found to host the antibacterial activity at MIC = 128 µg/mL against both bacteria, whereas no or little activity was noted in the petroleum ether and chloroform fractions (Figure 1).

Subfractionation of the aqueous subfraction by means of silica gel open-column and RP flash chromatography achieved a subfraction 3.1, showing a single spot on a reversed-phase TLC plate (Rf = 0.72; n-hexane: ethyl acetate: water, 5:13:2 *v*/*v*/*v*) at a yield of 34.2% *w*/*w* (Figure 1). Despite a fourfold increase in the antibacterial activity of the purified water fraction (MIC = 32 µg/mL), HPLC analysis of subfraction 3.1 showed multiple peaks under the UV chromatogram (RT = 6.0–7.8 min), denoting the presence of multiple closely related compounds. Further analysis of subfraction 3.1 under preparative RP-HPLC conditions (acetonitrile: water, 20–30%, 25 min. gradient) revealed at least 15 UV peaks (254 nm) of broadly varying sizes and retention times (Figure 1a). The subfractions corresponding to the observed peaks were collected, as indicated in Figure 1a, all first minor fractions were pooled in one fraction, and the five major subfractions (A–E) were individually collected (Figure 1a–e and Appendix A).

Antibacterial activities in the range of 32–128 µg/mL were observed towards the reference *E. coli* and *K. pneumoniae* strains in the subfractions corresponding to all the major peaks (A–E), in which fractions A and E exhibited the highest and lowest activities, respectively (Figure 1). As shown in Figure 1, the purified and semipurified compounds from subfractions A–E exhibited varying levels of antibacterial activities against reference and MDR strains of *E. coli* and *K. pneumoniae*. The active compounds corresponding to UV peaks/subfractions A2, B4, B6, and C3 were obtained in substantial quantities, and purity levels and were selected for structural elucidation and other biological studies. Further purification approaches to separate the compounds in subfraction D 3 and 4 under subfraction D were unsuccessful.

### 3.2. Characterization and Structural Elucidations of Selected Isolated Compounds

The infrared spectra of compounds corresponding to peaks A2, B4, B6, and C3 were similar and largely superimposable. Characteristic absorptions across the four compounds corresponded to aromatic O–H stretching (3353–3383 cm^−1^), aryl-substituted ester C=O stretching (1698–1704 cm^−1^), aromatic C=C stretching (1609–1610 cm^−1^ and 1446–1448 cm^−1^), alcoholic ester C–O stretching (1313–1316 cm^−1^), phenolic C–O stretching (1191–1195 cm^−1^), and ester C–O stretching (1026–1029 cm^−1^) (Appendix A).

These data, together with the corresponding HR-ES-MS, 1D, and 2D NMR spectra and comparison with data from spectral databases and the literature, enabled the identification of the compounds, as indicated below.

The compound corresponding to UV peak A2 in Figure 1b was obtained as an amorphous white powder and was assigned the molecular formula of C_41_H_32_O_26_ by its HR-ESI-MS *m/z* of 963.1054 [M+Na]^+^ (calculated for 963.1074). The compound was identified as 1,2,3,4,6-Penta-O-galloyl-β-d-glucose (PGG) and was coded as NMA2 (Figure 2 and Appendix A; Table 1) [23].

The compound corresponding to UV peak B4 in Figure 1d was obtained as an amorphous white powder and was assigned a molecular formula of C_48_H_36_O_30_ by its HR-ESI-MS *m*/*z* of 1115.1166 [M+Na] ^+^ (calculated for 1115.1184). The compound was identified as 3-O-digalloyl-1,2,4,6-tetra-O-galloyl-β-d-glucose and was coded as NMB4 (Figure 2 and Appendix A; Table 1) [22].

The compound corresponding to UV peak B6 in Figure 1c was obtained as an amorphous white powder and was assigned a molecular formula of C_48_H_36_O_30_ by its HR-ESI-MS *m*/*z* of 1115.1167 [M+Na] ^+^ (calculated for 1115.1184). The compound was identified as 6-O-digalloyl-1,2,3,4-tetra-O-galloyl-β-d-glucose and was coded as NMB6 (Figure 2 and Appendix A; Table 2) [24].

The compound corresponding to UV peak C3 was obtained as an amorphous white powder and was assigned a molecular formula of C_55_H_40_O_34_ by its HR-ESI-MS *m/z* of 1267.1268 [M+Na] ^+^ (calculated for 1267.1293). The compound was identified as 3,6-bis-O-digalloyl-1,2,4-tri-O-galloyl-β-d-glucose and was coded as NMC3 (Figure 2 and Appendix A; Table 2) [25].

The isotopic distributions observed in HR-ESI-MS for all four compounds were consistent with the assigned molecular formula, as it was also revealed by the spectra simulated from the respective molecular formula (Appendix A).

### 3.3. Antibacterial Activity Testing

Table 3 shows the MIC values obtained from broth microdilution susceptibility testing of compounds NMA2, NMB4, NMB6, and NMC3 against both the reference strains of *E. coli* and *K. pneumoniae* and those showing multidrug resistance to the indicated antibiotics. The observed MIC values of the tested compounds were noted to vary depending on the type of compound, as well as the prevailing resistance phenotypes of the studied bacteria. Upon MBC testing, bacterial growth colonies on agar plates were observed in wells containing compound concentrations greater than four times the MIC values; therefore, the compounds were regarded as bacteriostatic.

### 3.4. In Vivo Antibacterial Assays

At an inoculation dose of 1.5 × 10^8^ CFU/mL, the 24 h untreated *G. mellonella* larvae survival probabilities were 80% for *K. pneumoniae* (ATCC 10031) and 90% for *E. coli* (ATCC 25922)-infected larvae. This dose was therefore used to inoculate larvae in the subsequent tests for in vivo antibacterial activities of compounds NMA2, NMB4, NMB6, and NMC3.

A higher probability of survival was observed among larvae treated with the test compounds as compared to the negative controls after 96 h of incubation. Whereas the probability of survival among the *E. coli*-infected larvae was in the range of 20–40% (negative control = 5%), the larvae infected with *K. pneumoniae* showed 40–65% survival probabilities (negative control = 10%). The probabilities of survival of the larvae treated with gentamicin as a positive control were 20% and 45% among the *E. coli* and *K. pneumoniae*-infected larvae, respectively (Figure 3 and Figure 4).

### 3.5. Relationships between the MIC Values and the Nature of Expressed Resistance Enzymes by the Tested MDR Strains

Beta-lactamases formed a majority of the enzymes indicated to be expressed by the tested MDR strains of *E. coli* and *K. pneumoniae*. The relative contents of aromatic amino acids among the resistance enzymes expressed by the tested MDR bacteria were found to be in the range of 7.02–15.90% *w*/*w*. Moreover, the calculated net charges within these enzymes ranged from −15 to 4 (Table 4).

We observed a pattern of relationships between the magnitude of the observed MIC values and the content of the aromatic amino acids, as well as the net charge of the resistance enzymes expressed by the tested MDR strains. MIC values of 32 μg/mL or lower were observed among the bacteria expressing resistance enzymes with more than 11% *w*/*w* of aromatic amino acids or those with a net-zero or positive charge (Table 4, Figure 5).

These observations prompted a hypothesis with respect to the possible role of the expressed enzymes in influencing the ultimate susceptibility of the respective bacteria strains to the tested compounds. Figure 6 shows the possible interplay between bacteria, enzymes of varying nature, and the studied compounds. Enzymes expressed with higher proportions of aromatic amino acids and/or zero or positive net charge might cause increased proximity, higher surrounding concentrations, and enhanced interactions of the test compounds with the outer bacterial cell membrane (Figure 6A). Collectively, these factors could result in higher susceptibilities of the respective bacteria to the tested compounds. On the contrary, enzymes with contrasting features might bring about lower concentrations of the test compounds around the bacterial cells, leading to lower susceptibilities in such cases (Figure 6B).

## 4. Discussion

Extraction, isolation, and purification of bioactive compounds from crude plant extracts is an acceptably challenging task. This is mostly due to the complexity of a majority of crude plant mixtures, posing difficulty in the hunt for compounds exhibiting the activities of interest. The choice of acetone as an extractant was guided by previously reported antibacterial activities of *Paeonia broteoli* leaf extracts obtained from multiple solvents with varying polarities [17,38]. Other findings have shown the type and molecular weight of galloylglucoses recovered during extraction to be highly dependent on the nature of the extractant used. Acetone and ethyl acetate are the most suitable solvents for the extraction of low-molecular-weight galloylglucoses, among other tannins. Conversely, methanol and other organic solvent–water mixtures were found to mostly recover higher-molecular-weight galloylglucoses [10]. Although the nature of the antibacterial compounds was not known at the time of extraction, avoiding the use of acids to modify the extracting solvent and the isolation mobile phases was crucial because acidic conditions would have encouraged the hydrolysis of galloylglucoses in the course of their extraction and isolation [39].

Through contact bioautography, the identification of fractions/spots containing compounds with antibacterial activity was prominently simplified (Figure 1). The outcomes of bioautographic screening considerably influenced subsequent focus and choice of other isolation techniques to be employed. Whereas three main bioautography techniques are known, the type of bacteria under study can influence the outcomes of each technique [18]. Here, both *E. coli* (ATCC 25922) and *K. pneumoniae* (ATCC10031) performed better with contact bioautography as compared to direct TLC and immersion/agar overlay bioautography techniques.

The presence of multiple subfractions (A–E) with antibacterial activities from semipurified aqueous subfraction 3.1 signaled the possible existence of numerous structurally similar compounds in this subfraction (Figure 1a). This was underscored by the findings of at least two UV peaks (compounds) within each of subfractions A–E (Figure 1; Figure 1b–e; Appendix A). As a consequence, those mixtures required the use of isolation techniques and methods with a higher-resolution power and, in most cases, longer isolation times. The use of separate methods customized for the isolation of compounds within each subfraction (A–E) was crucial for the final step, in which compounds with suitable levels of purity were isolated.

Priority for further subfractionation was given to compounds isolated under peaks A2, B4, B6, and C3, in which suitable activity levels and complete isolations in appropriate quantities were attained (Figure 1, Figure 1b–e). Moreover, the efforts to purify subfractions D3 and D4 were not successful (Figure 1, Appendix A), and subfractions E1–E5 were not further characterized due to low or lack of antibacterial activities (Figure 1, Appendix A).

The presence of many structural isomers of galloylglucoses in the studied extract challenged the isolation of the compounds exhibiting antibacterial activities. The same challenge was previously implicated in studies involving a similar type of compound [21]. Owing to the limited number of studies conducted using purified galloylglucoses of known structures, reports on methods for their isolation and purification are valuable.

All of the isolated compounds were previously isolated and characterized from several other plant species. To the best of our knowledge, the isolation of compounds NMB4, NMB6, and NMC3 from *P. officinalis*, as well as their in vitro and in vivo antibacterial activities against reference and MDR strains of *E. coli* and *K. pneumoniae*, are reported here for the first time.

The spectrometric and spectroscopic data presented herein are similar to those previously reported for the same compounds isolated from other plant species [13,20,21,22,40]. Among others, the resonances typical of the glucose moiety are those corresponding to the anomeric carbon (C-1) appearing at δ = 93.11–93.40 ppm across all compounds (Table 1 and Table 2). The chemical shifts are indicative of the presence of beta-d-glucopyranose anomers, in contrast to the alpha anomers (δ~90 ppm) [41]. Additionally, the beta anomers of galloylglucoses are most commonly isolated from nature, whereas the occurrence and isolation of alpha anomers is very rare [12,42]. Moreover, the characteristic splitting of the ^1^H NMR shifts corresponding to methylene C-6 of the glucose core was noted at δ = 4.29–4.58 ppm in pentagalloylglucose (NMA2) as a split duplet of duplets. Signals in a similar ppm range were also evident in the remaining compounds. The existence of a methylene group at this position was confirmed by the ^13^C DEPT-135 spectra, showing δ = 62.80–63.43 across all compounds (Table 1 and Table 2; Appendix A).

The galloyl units surrounding the glucose core were characterized by the resonances of the protons at position 9 of the galloyl groups (δ = 6.92–7.51), among other signals. Signals resulting from these protons appeared as five distinct singlets in compound NMA2 and were more complex among the hexa- and heptagalloylglucoses (Table 1 and Table 2; Appendix A). The resonances conforming to the carbonyl carbons on the ester groups of the galloyl units occurred at δ = 165–166.87 ppm across all compounds (Table 1 and Table 2). Although galloyl units are typically esterified with a glucose polyol to yield galloylglucoses such as those reported here, other possible polyols include fructose, saccharose, xylulose, glucitol, and hamamelose, as well as shikimic and quinic acids [10,39].

The ascertainment of the exact position of attachment of each galloyl unit (A–E) on carbons C 1–4 and C–6 of the glucose cores was a key undertaking. This was achieved by aligning the three-bond correlations (HMBC) between each of the protons on the glucose core and the carbonyl carbons at positions 7a–7e on one hand, as well as between those carbonyl carbons and the aromatic protons at positions 9a–9e on the galloyl units on the other hand (Table 1 and Table 2; Figure 2 and Appendix A). Following this step, the determination of the galloyl units carrying an additional/distal galloyl unit (digalloyl units) was feasible.

The attachment of a (distal) galloyl unit on a particular galloyl group proximal to the glucose core by a depsidic bond resulted in the downfield shift of the ^13^C and ^1^H signals originating from the respective proximal galloyl unit (Table 1 and Table 2). The resulting most downfield ^1^H signals on the aromatic region corresponding to those at position 9 of the proximal unit of the digalloyl moieties were therefore earmarked. This was supported by the convergence of the HMBC correlations of such protons and those of the protons in the respective position on the glucose core to the same carbonyl carbon (position-7) of the ester bond in between (Table 1 and Table 2; Figure 2).

Despite the three-bond HMBC correlations between the protons at positions 9c’ (NMB4), 9e′ (NMB6), as well as those at positions 9c′ and 9e′ (NMC3) and the carbonyl carbon on the adjacent depsidic bonds, the lack of protons in proximity on the proximal galloyl unit hindered the determination of connectivity using HMBC alone (Figure 2). Nevertheless, due to the deshielding effect of the proximal galloyl unit on the distal galloyl unit, the shifts resulting from protons at position 9 of the distal galloyl units appeared at the second most downfield positions in the aromatic region of the spectra and correlated with the carbonyl carbon in the depsidic bond rather than that in the underlying ester bond (Table 1 and Table 2; Appendix A) [43]

Other authors have frequently indicated the use of the downfield shifting of the ^13^C NMR resonances of the respective carbons on the glucose core in comparison to those of the pentagalloylglucose to justify the position of the distal galloyl groups on hexa-, hepta-, octa-, and other polygalloylglucoses [20,21,41]. Similarly, in compound NMB4, the downfield shift of the C-3 signal led to its overlap with the C-5 signal at δ = 73.44 ppm. The same phenomenon was noted among the signals due to C-6 in NMB6 and C-3 plus C-6 in NMC3 (Table 1 and Table 2) [21]. Using a similar approach, Nishizawa and Yamagishi implied that the C-3 and C-6 positions of the glucose core are predominant for attachments of depsidic galloyl groups [21].

Previous studies have indicated that in solutions, the distal galloyl groups tend to migrate between the ortho and para positions of the proximal galloyl unit and coexist in an equilibrium mixture of the two isomers [20]. Furthermore, it was noted that this migration induces the shifting of the ^1^H and ^13^C NMR resonances in other positions of the respective compounds, resulting in multiplets and an increased overlapping pattern of the signals (Table 1 and Table 2; Figure 2) [21]. In addition, the stated migration causes the observed differences in resonances of the protons attached at position 9 of the proximal galloyl units. Therefore, the carbon (position 9) next to the meta carbon appears to be more downfield-shifted (^13^C δ~117 ppm) when the distal galloyl group has migrated to the meta position, whereas the corresponding carbon in a similar position resonates at δ~114 ppm. Furthermore, the shifts of the protons in these positions appeared to follow the same pattern. On the other hand, the migration of the distal galloyl unit to the para position resulted in unified and more upfield-shifted (^13^C δ = 109–110 ppm) signals corresponding to position 9 of the proximal galloyl unit (Table 1 and Table 2). Due to the observed migratory nature of the distal galloyl groups in compounds NMB4, NMB6, and NMC3 in solution, it was not possible to ascertain their exact position(s) (meta, para, or a mixture of both) in these compounds based on NMR data alone.

The isolated galloylglucoses showed bacteriostatic activities against *E. coli* and *K. pneumoniae* strains of different resistance phenotypes. Generally, some or all compounds exhibited higher activity against MDR strains expressing KPC-2, KPC-3, OXA-48, and DHA-1 enzymes. Conversely, moderate or lower activity levels were observed among the strains expressing VIM-1, VIM-4, TEM-1, SHV, MCR-1, and NDM-1, either alone or together with other enzymes. The MIC values observed for *E. coli* and *K. pneumoniae* strains with ESBL and DHA-1 phenotypes, respectively, were many folds lower among the compounds NMB6 and NMC3.

The observed differences in the susceptibilities of the studied MDR bacteria to galloylglucoses are unlikely to be based on enzyme–substrate interactions due to the broad structural differences between galloylglucoses and the usual substrates (e.g., beta-lactam antibiotics) of the resistance enzymes expressed by the studied MDR bacteria. The influence of other resistance-enzyme-related factors might have therefore contributed to the observed variation in susceptibilities.

Antimicrobial activities of galloylglucoses have been reported in various species of bacteria, fungi, and viruses [10,15,42]. Similar to our findings, previously reported antibacterial activities were mainly bacteriostatic against both Gram-negative and Gram-positive bacteria, and the Gram-negative bacteria were less susceptible [10,44,45]. Moreover, similar antibacterial activities of pentagalloylglucose (MIC or minimum regrowth concentration (MRC) = 32–256 μg/mL) against different reference and MDR strains of *E. coli* and *K. pnemouniae* were previously reported [46,47,48]. Additionally, galloylglucoses are reported to inhibit extracellular bacterial enzymes, toxins, adhesins, surface transport proteins, and biofilm formation [10,11,14]. These activities signify the potential of galloylglucoses against different mechanisms of pathogenicity and antibacterial resistance. No antibacterial activities of compounds NMB4, NMB6, and NMC3 were previously reported. However, compound NMB4 was reported to inhibit the enzyme alpha-glucosidase, the influx of Ca^2+^ in skin and respiratory cells, lipid formation in adiposities, and Alzheimer’s amyloid beta-peptide aggregation [22,49,50,51]. Further, compound NMC3 was reported to block cell-membrane-based Ca^2+^-dependent-chloride currents, induce formation of interferon, and exhibit antitumor activity [29,52].

Very low solubility of the isolated compounds was observed in solvents systems commonly used for broth microdilution assays. Complete dissolution of the compounds could not be attained using up to 2.5% DMSO in water or MHB media. This prompted efforts to explore other solubilization approaches to enable a more objective screening of the compounds’ antibacterial potentials. This was achieved by preparing stock solutions by first dissolving the compounds in acetone, followed by working solutions, which contained 25.6% *v*/*v* of acetone in MHB. Ensuring a uniform concentration of acetone across all test wells (12.8% *v*/*v*) and filling the outermost wells with MHB media minimized acetone evaporation during the incubation time. Previous studies showed non-toxicity to bacteria at concentrations of up to 25% *v*/*v* of acetone in the test media [53,54].

The low water solubility of galloylglucoses hinders objective investigation of their antibacterial potentials in vitro and in vivo. Many studies have opted for disc diffusion assays, in which galloylglucoses are dissolved in an organic solvent before loading the discs. This approach achieves proper solubilization but is subject to less objective results, as the diffusion of compounds into water-based agar media is apparently low [16,44,55]. Furthermore, the solubility of galloylglucoses is highly influenced by the extent of their galloylation; those with more than four galloyl groups show lower water solubility profiles as compared to those with a lesser degree of galloylation [31]. This decrease in hydrophilicity is related to an increased degree of intramolecular hydrogen bonding and intermolecular stacking attained with a higher number of galloyl groups [39]. The degree of galloylation might therefore be important with respect to finetuning the balance between compound solubility in test media and the degree of lipophilicity ideal for their interaction with bacteria cells. This is emphasized by the occurrence of optimal antibacterial activities among galloylglucoses with 6–7 galloyl groups [11].

All compounds resulted in *G. mellonella* larvae survival rates similar to or higher than those of the positive control (gentamicin); compound NMC3 ensured the best survival rates of larvae against both *E. coli* and *K. pneumoniae*. Furthermore, all compounds yielded better survival rates among the larvae infected with *K. pneumoniae* than with *E. coli*, which was consistent with the in vitro profiles.

The availability of data on in vivo antibacterial activities of galloylglucoses in higher animals is limited by their low oral bioaccessibility and bioavailability levels [10,39]. Improved in vivo anticancer and antiallergy activities were observed when galloylglucoses were administered via intraperitoneal or intravenous routes [13,32]. Conversely, other researchers have questioned the in vivo activities of galloylglucoses based on their likelihood of interacting with numerous proteins, limiting the attainment of effective concentrations [56]. Moreover, galloylglucoses are substrates of a range of hydrolytic and oxidative enzymes produced by gut microbiota in higher animals [10,39]. Most of the resulting metabolites can be absorbed and are highly linked to the observed systemic activities after oral administration [39].

The strains expressing enzymes KPC-2, KPC-3, OXA-48, DHA-1, and CTX-M, which have 11.7–15.4% *w*/*w* aromatic amino acid content and 0–4 net charges, were more susceptible (MIC = 2–64 μg/mL) to at least two of the galloylglucoses (Figure 5; Table 3). On the other hand, strains expressing VIM-1, TEM-1, SHV-1, and NDM-1 with aromatic amino acid contents of 7.0–9.5% *w*/*w* only and net charges of −1 to −15 were generally less susceptible (MIC = 64– > 256 μg/mL) to all galloylglucoses. Furthermore, the *E. coli* strains with phenotypes for both KPC-2 (net charge = +1) and MCR-1 (net charge = −12) were the least susceptible to each of the galloylglucoses (MIC > 256 μg/mL) (Figure 5; Table 3). These findings suggest a relationship between the nature of the resistance enzymes expressed by the MDR bacteria and their susceptibility to galloylglucoses.

However, the MIC values observed in the strain of *E. coli* expressing resistance enzyme OXA-48 (net charge = 0, aromatic AAs content = 15.4% *w*/*w*) were remarkably higher (256- > 256 μg/mL) than those in the *K. pneumoniae* strain expressing the same enzyme (4–32 μg/mL) (Table 3). Thus, we postulate that other factors, such as the presence of unidentified resistance enzyme(s) with opposing features, favored the observed lower susceptibility of the *E. coli* strain.

The observed antibacterial activities of galloylglucoses are related to their previously reported ability to interact with proteins, carbohydrates, lipids, and metal ions [10,11]. The compounds characteristically bind to different macromolecules through hydrophobic interactions, as well as via hydrogen, covalent, and ionic or electrostatic bonds [10,15,39]. Proteins with higher proportions of aromatic amino acids were reported to show stronger hydrophobic interactions with galloylglucoses [29,30,31,33]. The compounds are also capable of electrostatically adsorbing to surfaces of macromolecules or elements carrying opposite charges [15,29,32,34].

These behaviors might explain the observed variations in MICs of the investigated compounds among MDR strains expressing enzymes with different contents of aromatic amino acids and net charges. The nature of resistance enzymes might influence the ultimate concentration of galloylglucoses around bacterial cells. To this end, enzymes richer in aromatic amino acids or with zero or positive net charges attract and interact more with galloylglucoses. The presence of those enzymes and their interactions with galloylglucoses might result in higher concentrations of galloylglucoses in the vicinity of bacterial cells (Figure 6A). Therefore, the compounds can attack the bacterial cells more intensely via a number of previously described modes of action. In contrast, the presence of enzymes with lower content of aromatic amino acids and/or net negative charges can accomplish the opposite effect [15,30,31,32,33,34]. In this case, lower concentrations of galloylglucoses around the bacterial cells make the respective bacteria less susceptible (Figure 6B).

## 5. Conclusions

Screening and isolation of antibacterial compounds from nature remains an important and challenging approach to the discovery and development of novel antibiotics. This study highlights a range of useful approaches to first-time extraction, isolation, purification, and characterization of three of the four galloylglucoses from the leaves of *P. officinalis*. Importantly, the challenge posed by the common coexistence of closely related galloylglucoses was mostly addressed by a combination of bioautography-guided extractive and chromatographic techniques.

The observed moderate-to-high bacteriostatic activities of the isolated compound against reference and MDR strains of *E. coli* and *K. pneumoniae* underline the previous reports on antimicrobial activities of galloylglucoses. Furthermore, the relative content of aromatic amino acids and net charges of the expressed resistance enzymes were noted to influence bacterial susceptibilities to the studied galloylglucoses. Moreover, diverse modes of action targeting different macromolecules on bacterial surfaces, as well as enzymes, toxins, and nutrients in the surrounding media, were previously indicated.

Despite limitations with respect to their absorption, metabolism, and lower target selectivity, galloylglucoses can potentially be applied in the agriculture and food industries, as well as in the management of septic wounds and other topical microbial infections. Through these and other possible avenues, galloylglucoses can substantially contribute to supplementing, reducing, or replacing the use of contemporary antibiotics in order to mitigate the development of antimicrobial resistance.

## Data Availability

The data presented in this study are available in the article and the supporting information.

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
