# Peer review of "Isolation and Characterization of Galloylglucoses Effective against Multidrug-Resistant Strains of Escherichia coli and Klebsiella pneumoniae"

_molecules, 2022, doi:10.3390/molecules27155045_

Round 1
Reviewer 1 Report
The authors reported the isolation and characterization of four galloylglucoses active against different MDR strains of E. coli and K. pneumoniae. The compounds was isolation by liquid-liquid extraction, open column-, flash- and preparative chromatographic methods, and then characterized FT-IR, HR-ESI-MS and NMR spectroscopic techniques. Various bioactivities of isolated compounds were conducted. The whole paper is well organized and could be revised for better reading and understanding.
l Table 2 and Table 3 could be put into the supplementary files.
l Author could give a summary information of isolated compounds.
l The conclusion part did not showed the highlights of this research.
l Author could give a evaluation of the activity of isolated compounds and make a comparison.
Reviewer 2 Report
The manuscript entitled '' Isolation and Characterization of Galloylglucoses Effective against Multidrug-Resistant Strains of Escherichia coli and Klebsiella pneumoniae'' is very interesting from the medicinal viewpoint; however, the authors need to make some changes.
· In the Abstract section,
- Abbreviations such as MDR should be written words as it first appears. So, check the abbreviations throughout the manuscript. It would be best to introduce the abbreviation when the whole word appears the first time in the text and then use only the abbreviation.
- Please write if the isolation and characterization of four galloylglucoses for the first time.
- Please complete the sentence in Line 18.
- Paeonia officinalis should be written with its authority name at least in the first appearance and plant material section.
· In the Materials and Method section
- Please, add the yield of crude extract in line 118.
- Please, transfer Table 1 into Supplementary data.
- Please, delete the repeated sentence in line 219, ''Broth Microdilution Assays for MIC Determination''.
· In the Result section
- Where is the Figure 1?
- It is better to collect all UV chromatograms in one Figure to be, for example, Figure 1 A, B, C, and D
- In Tables 2 and 3, remove spaces before and after to present the data in a good form.
· In the Discussion section
- No need for subsection titles.
- Please summarize the Discussion part as it was wordy. The authors should make it concise and interesting for readers not to be too long or boring. Also, the authors should not repeat the Results Data.
· Some grammatical, alignment and typographical errors are noted in the manuscript, and they should be thoroughly checked and corrected throughout the manuscript.
· Please carefully check the numbering of tables and figures.
· Finally, the authors present a comprehensive study and good efforts in the manuscript, but it still needs some reorganization and summarization in some parts to be more concise. Also, transfer unnecessary or raw data into the Supplementary file.
Reviewer 3 Report
· The authors have performed the isolation and characterization of four galloylglucoses against different E. coli and K. pneumoniae. Distinct chromatographic techniques were used to extract and isolate the active compounds as well as spectroscopic and spectrometric techniques to characterize them. Additionally, it was demonstrated their potential against MDR gram-negative bacteria. Based on these exhaustive and detailed manuscript, I recommend for publication.
· Minor Suggestions:
· Characterization: the isotopic distribution in the HRMS analyses should be mentioned if are in agreement with the molecular formula.
· Characterization: in the HMBC NMR experiment, the long-distant parameter (8, 10 or 12 Hz) should be described in this section.
· Results: the chromatograms should be moved for the support information.
· Discussion: the yields for extraction, isolation and purification could be compared with the literature.
Round 2
Reviewer 2 Report
Good efforts